# Theoretical Insight into Psittacofulvins and Their Derivatives

**DOI:** 10.3390/molecules29122760

**Published:** 2024-06-10

**Authors:** Marcin Molski

**Affiliations:** Department of Quantum Chemistry, Faculty of Chemistry, Adam Mickiewicz University of Poznań, ul. Uniwersytetu Poznańskiego 8, 61-614 Poznań, Poland; mamolski@amu.edu.pl

**Keywords:** polyenes, psittacofulvins, thermodynamic descriptors, chemical activity descriptors, DFT method

## Abstract

Psittacofulvins are polyenal dyes responsible for coloring parrot feathers and protecting them against photo-oxidation, harmful radicals, and bacterial degradation. To explain the unusual properties of these compounds, the thermodynamic and global chemical activity descriptors characterizing four natural and three synthetic psittacofulvins, as well as their hydroxyl, carboxyl and dialdehyde derivatives, were determined. To this aim, the DFT method at the B3LYP/QZVP theory level and the C-PCM solvation model were used. The calculations enabled the selection of the projected compounds for the greatest bioactivity and potential applicability as multifunctional ingredients in medicines, cosmetics, supplements, and food, in which they may play a triple role as preservative, radical scavenger, and coloring agent. The results obtained provide arguments for the identification of a fifth psittacofulvin within the parrot feather pigment, characterized by ten conjugated double bonds (docosadecaenal).

## 1. Introduction

Parrots are extraordinary birds distinguished by high intelligence, resistance to diseases, long lifespan, and the spectacular colors of their plumage. In 1882, Krukenberg discovered [1] that the pigment in the feathers of parrots contains unique lipochromes, called psittacofulvins or parrodienes, that do not belong to the classes of carotenoids, melanin, porphyrins and pterins usually responsible for the plumage of birds. Modern analytical techniques based on high-performance liquid chromatography (HPLC), coupled with visible and ultraviolet spectroscopy (UV/VIS), have enabled Stradi et al. [2] to identify the following four polyenal pigments in the feathers of parrots (see Figure 1): tetradecahexenal (P6), hexadecaheptenal (P7), octadecaoctenal (P8) and eicosanonenal (P9). 

They are characterized by a retention time t_R_ ranging between 5.5 and 7.5 [min], the electronic spectra with the absorbance maxima λ_max_ between 400 and 450 [nm], and the molecular masses 200, 226, 252, 278 [Da], respectively. McGraw and Nogare [3] confirmed that the pigments found in the red plumage of 44 species of parrots contain four dyes previously identified [2] with λ_max_ between 424 and 443 [nm] and a fifth, more polar, as-yet unidentified lipochrome with λ_max_ = 438 [nm] and t_R_ = 6.4. Spectroscopic studies on these pigments were also performed by Tay et al. [4] by making use of resonance Raman spectroscopy in situ. The investigations confirmed [4] the presence of dye molecules P6–9 as well as that of an additional one with a greater conjugation of bonds than that reported for P9. The optical characteristics of the identified compounds contrast with empirical observations, as light absorption in the range 400–450 [nm] results only in the yellow-green (380–435 [nm]) or a yellow (435–480 [nm]) complementary (perceived) color of parrot feathers, whereas parrot plumage is known to display a range of colors that furthermore includes red, orange, lilac, blue, purple, and indigo [2].This surprising effect has been explained [5] by the fact that electronic S_0_ − S_1_ transitions in polyenes are forbidden by quantum-mechanical selection rules, as both states belong to the same A_g_^−^ symmetry. Therefore, only the transitions to and from S_0_ − S_2_ (B_u_^+^ symmetry) states, responsible for the absorption of light in the narrow range 400 − 450 [nm], are allowed. However, in psittacofulvins, a polyene chain is terminated by an aldehyde group, influencing the ordering of states and the dynamics of the π-electron conjugation, including the carbonyl group. In such circumstances, the solvent-dependent intramolecular charge transfer (ICT) state can influence the S_1_ lifetime and generate S_1_→S_n_ transient absorption (TA). This results in the light reflectance and/or fluorescence responsible for the plumage of parrots on the molecular level. It was also hypothesized [2] that parrots create a wide spectrum of colors by modulating the interactions of the four (five) endogenous yellow pigments with plumage keratin. The latest findings [6], supported by genetic investigations [7,8,9,10], revealed that psittacofulvins are responsible for the yellow and red (via ICT and TA) coloration, while the blue color is the result of light being scattered by keratin nanostructures and melanin granules. This indicates that the vivid coloring of parrot feathers is the result of light interacting not only with the bio-pigments independent of keratin, but also the microstructural arrangement of feather tissues.

Parallel to the research aimed at explaining the nature of parrot plumage, a hypothesis was verified showing that psittacofulvins are responsible for the longevity of parrots (unique among birds) and their strong resistance to viral and bacterial infections. While not formally confirmed, the oldest parrots are said to have reached upwards of 80 years of age at the moment of their deaths in captivity, with some wild parrots often exceeding the 50-year mark. Since the aging process is influenced by, among others, exogenous and endogenous radicals that cause damage to DNA, RNA and lipids, psittacofulvins can be theorized to be endowed with high anti-radical activity. Theoretical research in this regard conducted by Martinez [11] confirmed that the pigments found in parrot feathers are effective scavengers of free radicals; however, they are not anti-oxidating, but rather, anti-reducing agents. Unfortunately, the chemical activity and thermodynamic descriptors that would explain the antiradical properties of psittacofulvins have not yet been determined. The investigations of parrot pigments performed by Burtt et al. [12] showed that they preserve plumage integrity by increasing the resistance of feather keratin to bacterial degradation. When red feathers were exposed to the feather-degrading *Bacillus licheniformis*, they were destroyed more slowly than those without pigments. In the case of blue and green feathers, which combine structural blue with yellow psittacofulvins, the rates of degradation were similar to those of red feathers [12]. The complementary experiments showed [13] that psittacofulvins and their short-chain derivatives [14,15,16] can play an important role in the inhibition and prevention of pathological disorders. In particular, all-trans-2,4-hexadienal, all-trans-2,4,6-octatrienal, and all-trans-2,4,6,8,10-dodecapentaenal were tested [16] for their antiproliferative activity against SH-SY5Y cells (thrice-subcloned cell line derived from the SK-N-SH neuroblastoma line) employed as a model for neurodegenerative diseases. Investigations revealed that 100 [µM] of trans-2,4-hexadienal significantly inhibited the growth of SH-SY5Y cells after 72 [h] of incubation, whereas all-trans-2,4,6-octatrienal and all-trans-2,4,6,8,10-dodecapentaenal were more effective at a lower concentration of 10 [µM]. Additionally, the latter compound, at a concentration of 1 [µM], protected 2-deoxyribose from the degradation caused by ferrous ions. The corresponding polyenol derivatives were less effective in this regard [16]. The results obtained prove [14,15,16,17] that the bioactivity of the compounds analyzed increases with the number of coupled double bonds enriched by the π-electrons from the carbonyl group. The application of the electron paramagnetic resonance (EPR) technique demonstrated [17] that these novel derivatives, especially octatriene, inhibit the formation of reactive oxygen species (the hydroxyl radical in particular), which are a significant source of damage to DNA, RNA, lipids, cells, and tissues. It is expected that polyunsaturated psittacofulvins should feature greater bioactivity than the compounds considered to date [14,15,16,17]. In view of this, the main purpose of the present study is to determine the thermodynamic and global chemical activity descriptors characterizing naturally occurring psittacofulvins and their hydroxyl, carboxyl, and dialdehyde derivatives presented in Figure 2. 

They will enable the selection of the projected compounds in terms of their greatest bioactivity and potential applicability. The PubMed database (keyword: psittacofulvins) indexes only 20 studies in the literature, including only one [11] on the theoretical determination of the reactivity descriptors by quantum–chemical calculations. For this reason, the present work is the first attempt to determine a wide range of activity parameters of the compounds considered, which will enable an explanation of the specific bioreactivity and uniqueness of psittacofulvins among other pigments utilized by animals and plants.

## 2. Computational Details

To achieve the main objectives of the present work, the following parameters characterizing the chemical and thermodynamic reactivity of psittacofulvins, and their derivatives, were determined:(i)the global descriptors of the chemical activity: the ionization potential IP, electron affinity EA, chemical potential μ, absolute electronegativity χ, molecular hardness η and softness S, electrophilicity index ω, the electro-donating ω^−^ and electro-accepting ω^+^ powers, and the Ra, Rd indexes [18,19,20];(ii)the thermodynamic descriptors: the bond dissociation enthalpy BDE, adiabatic ionization potential AIP, proton dissociation enthalpy PDE, proton affinity PA, electron transfer enthalpy ETE, the free Gibbs acidity H_acidity_ (in the gas phase) or G_acidity_ (in hydrophobic, i.e., benzene, and hydrophilic, i.e., water, solvents) [18,19,20].

The thermodynamic parameters characterize the radical X^●^ scavenging potency of the compound R–H, in which the hydrogen is removed from the aldehyde, carboxyl, or hydroxyl group. This reaction is characterized by the following deactivation mechanisms [18,19,20]:

**HAT**—(hydrogen atom transfer) associated with BDE, as follows:R–H + X^●^ → R^●^ + X–HBDE = H(R^●^) + H(H^●^) − H(R–H)

**SET-PT**—(single electron transfer followed by proton transfer) characterized by AIP and PDE, as follows: R–H + X^●^ → R–H^●+^ + X^−^ R–H^●+^ → R^●^ + H^+^ AIP = H(R–H^●+^) + H(e^−^) − H(R–H) PDE = H(R^●^) + H(H^+^) − H(R–H^●+^)

**SPLET**—(sequential proton loss electron transfer) described by PA and ETE, as follows: R–H → R^−^ + H^+^ R^−^ + X^●^ + H^+^ → R^●^ + X–H PA = H(R^−^) + H(H^+^) − H(R–H) ETE = H(R^●^) + H(e^−^) − H(R^−^)

**TMC**—(transition metal chelation) described by H_acidity_ (gas phase) and G_acidity_ (solvent); they characterize a tendency to chelate transition metal ions (especially Fe^2+^ and Cu^2+^), which may participate in the creation of radicals. Vital for this mechanism is the dissociation decay, as follows:R–H → R^−^ + H^+^
which produces the anionic form of R–H, active in TMC. The parameter characterizing TMC in solvents is the free Gibbs acidity defined by the Gibbs free energy of the anion G(R^−^) and its parent compound G(R–H), as follows: G_acidity_ = G(R^−^) − G(R–H) 

Here, H(R–H^●+^), H(R^●^), H(R^−^), and H(R–H) represent the enthalpies of the cation, radical, anion, and the parent compound, whereas H(H^●^), H(e^−^), and H(H^+^) are the enthalpies of the hydrogen, electron, and proton. A small value of the calculated parameter indicates a low energy of activation—the energy associated with dehydrogenation (HAT), ionization (SET-PT), and deprotonation (SPLET) in the initial stage of radical deactivation. In the case of two-stage processes, the sum of parameters (PA + ETE or AIP + PDE) should also be included. The global descriptors of chemical activity are defined by the HOMO and LUMO energies and their exact mathematical formulas are presented in [18,19] and the Appendix A. In the calculations, we used the following values (in Hartree [Ha] unit) of the electron, proton and hydrogen enthalpies in the gas phase [20]: H(e^−^) = 0.001198, H(H^+^) = 0.002363, H(H^●^) = −0.497640; water: H(e^−^)_aq_ = −0.03879545, H(H^+^)_aq_ = −0.38690958, H(H^●^)_aq_ = −0.49916356 and benzene: H(e^−^)_be_ = −0.00146823, H(H^+^)_be_ = −0.33815566, H(H^●^)_be_ = −0.495202304. The last six values can be calculated using the following relationships:H(e^−^)_so l_ = H(e^−^) + ∆_sol_H(e^−^), H(H^+^)_sol_ = H(H^+^) + ∆_sol_H(H^+^), H(H^●^)_sol_ = H(H^●^) + ∆_sol_H(H^●^)
in which ([kJ mol^−1^] unit) ∆_aq_H(e^−^) = −105, ∆_aq_H(H^+^) = −1022, ∆_aq_H(H^●^) = −4.0, ∆_be_H(e^−^) = −7.0, ∆_be_H(H^+^) = −894, ∆_be_H(H^●^) = 6.4 are the solvation corrections recommended by Rimarčik et al. [20]. To calculate H(R–H^●+^), H(R^●^), H(R^−^), H(R–H), and the HOMO–LUMO energies, we employed the DFT method implemented in Gaussian vs. 16 software [21]. The input structures were constructed by taking advantage of the Gauss View−6.1 graphical interface [21], whereas the calculations were carried out in the Supercomputing and Networking Center of Poznan. Since the computations for the compounds studied at the B3LYP/QZVP level are extremely time-consuming and their convergence depends on the initial geometry input, the optimization was divided into the following three stages: (i) the determination of an approximate geometry at the B3LYP/6311++G(d, p) level of the theory; (ii) the geometry refinement at the B3LYP/cc-pVQZ level; and (iii) the final calculation at the B3LYP/QZVP level of the theory. Thus, optimization was conducted at each stage, with the geometry determined at the lower level serving as the starting point for optimization at the higher level. The results obtained in this scheme are reported in Table 1, Table 2, Table 3 and Table 4 and displayed in Figure 3 and Figure 4. The final values of the descriptor were calculated using the Maple vs. 16 processor for symbolic computations.

## 3. Results and Discussion

The quantum–chemical calculations were divided into two parts. First, they were carried out only for P6 and its hydroxyl, carboxyl and dialdehyde derivatives (Figure 2) to select those compounds that show the highest reactivity in the gas phase as well as in the hydrophilic (water) and hydrophobic (benzene) media. In the preliminary calculations performed for P6 in the gas phase, the total energy was calculated at the B3LYP/cc-pVQZ, B3LYP/aug-cc-pVQZ, and B3LYP/QZVP levels of the theory consisting of Becke’s [22] exchange functional in conjunction with the Lee–Yang–Parr [23] one and the correlation-consistent polarized-valence quadruple zeta basis sets (cc-pVQZ and aug-cc-pVQZ) introduced by Dunning [24] as well as the Weigend–Ahlrichs valence quadruple-zeta polarization basis set (QZVP) [25]. The results obtained for P6 in the gas phase: E(cc-pVQZ) = −618.547012, E(aug-cc-pVQZ) = −618.549867, E(QZVP) = −618.557811 [Ha] allowed for the selection of the optimal basis set QZVP, which was used in the test calculations with other functionals [26,27] to provide the following energy values: E(M06-2X) = −618.261159, E(BHandHLYP) = −618.146620, E(LC-ωPBE) = −618.066651, E(LC-ωHPBE) = −618.065122 [Ha] in comparison to E(B3LYP) = −591.436087 [Ha]. The values of E(ccp-VQZ) = −618.547012 and E(aug-ccp-VQZ) = −618.549867 [Ha] indicate that the inclusion of diffusion correction in the aug-ccpVQZ basis set diminishes the total energy value by only 1.79 [kcal mol^−1^], while QZVP produces E(QZVP) = −618.557811 [Ha]—the difference is 6.78 [kcal mol^−1^]. Consequently, the geometry optimization and enthalpy calculations for all molecules considered were performed at the B3LYP/QZVP level of the theory as it produced the lowest total energy of P6. To investigate the effect of the functional type on the values of chemical activity parameters, calculations were also carried out for M06-2X and BHandHLYP—the functionals that ranked second and third in the P6 energy ranking reported above. The results of the calculations are presented in Appendix A. The calculations were also performed in hydrophilic (water) and hydrophobic (benzene) media by taking advantage of the C-PCM solvation model [28] (conductor-like polarizable continuum model). The application of other models resulted in greater energies E(IEF-PCM) = −618.570414 and E(SMD) = −618.569309 [Ha] in comparison to E(C-PCM) = −618.570497 [Ha]. Additionally, the calculations performed for gallic acid showed [19] that the B3LYP/C-PCM combination correctly reproduces the difference in HOMO−LUMO energies as well as the values of chemical activity parameters. Therefore, the C-PCM model appears as optimal in the calculation of chemical activity and thermodynamic descriptors for the molecules considered. The calculations performed for the P6-H_2_O system (B3LYP/QZVP) in the water medium (C-PCM) provided the values of the parameters that are slightly different from those obtained for a single P6 in an aqueous medium (C-PCM). P6-H_2_O: E(HOMO) = −0.20275 [Ha], E(LUMO) = −0.11036 [Ha], ∆E = 2.5141 [eV]; P6: E(HOMO) = −0.20128 [Ha], E(LUMO) = −0.10668 [Ha], ∆E = 2.5742 [eV]. Since the remaining chemical activity descriptors depend on ∆E, their values will also change slightly.

The results presented in Table 1 demonstrate the following:(i)the hydrophobic environment does not affect the antiradical activity of the tested compounds, which is comparable to that in a vacuum and characterized by the BDE parameter related to the HAT mechanism;(ii)in an aqueous medium, the antiradical activity of the derivatives analyzed increases, whereas the scavenging mechanism preferred changes to SPLET;(iii)the hydroxyl derivative POH6 of P6 shows less activity than the parent compound (the SPLET total energy PA + ETE = 145.27 [kcal mol^−1^]); therefore, it will not be considered at the second stage of calculations;(iv)D6 has (PA + ETE = 119.77 [kcal mol^−1^]) greater antiradical activity than P6, whereas POOH6 is a stronger metal chelation agent (G_acidity_(H_2_O) = 286.42 [kcal mol^−1^]) than P6; therefore, the research will also cover their polyunsaturated derivatives based on the P7–9 parent compounds.

In light of the points (i)–(iv), calculations of the activity descriptors of the selected compounds P4–10, POOH6–9, D6–9 in the water environment were carried out. The results presented in Table 2 show that the four naturally occurring psittacofulvins P6–9 and their modeled derivative P10 deactivate radicals according to the SPLET mechanisms. This activity depends on the number of conjugated bonds in the polyene chain and increases from P6 up to P10. P5 is an intriguing case as the difference PA − BDE = 0.75 [kcal mol^−1^] is less than the accuracy level required in realistic chemical predictions. Hence, one may assume that both HAT and SPLET mechanisms can be activated simultaneously during the deactivation of radicals, contributing equally to the hybrid mechanism of the radical scavenging activity. This trend is maintained; therefore, starting from P4, a change in the radical deactivation mechanism from SPLET to HAT takes place.

Comparative calculations revealed (Table 3 and Table 4) that the carboxyl and dialdehyde derivatives of P6–9 exclusively prefer the SPLET scenario. These results indicate that, in the process of evolution, parrots have developed an extremely effective defense system, which protects them against the devastating and pathogenic (especially teratogenic and carcinogenic) influence of radicals. Since psittacofulvins additionally exhibit antimicrobial activity [14,15,16,17], parrot feather dyes can be said to form a multifunctional complex of bioactive compounds that protect the animal against the microorganisms and exogenous radicals created by UV radiation.

The values of the global descriptors of the compounds studied reported in Table 2, Table 3 and Table 4, as well as in Appendix A indicate a systematic growth of chemical reactivity with respect to the number of conjugated double bonds. This effect is independent of the functional used in the calculations. The results obtained by taking advantage of the M06-2X and BHandHLYP functionals reveal that although the values of calculated descriptors differ from those obtained with B3LYP, the conclusions regarding the activity of the compounds considered remain consistent. In particular, for B3LYP and psittacofulvins P4−10 the energy gap ΔE = E(LUMO) − E(HOMO) decreases from 3.1954 (P4) to 1.9779 (P10) [eV]. A large HOMO–LUMO energy difference indicates a hard molecule that is more stable and less active, while a small energy gap characterizes a soft molecule that is less stable and more reactive. In the case of psittacofulvins and their derivatives, ΔE is extremely small. For comparison, for 2-propene sulfenic acid, one of the strongest known radical scavengers identified in garlic metabolites [18], ΔE = 5.6461 [eV], whereas for gallic acid, an extremely potent antioxidant, ΔE = 4.9625 [eV] [19]. Generally, the following ranking ΔE(POOHx) > ΔE(Px) > ΔE(Dx) for x = 6,7,8,9, of the energy gaps can be indicated. The smaller value of IP = 5.0572 [eV] for P10 in comparison to IP = 5.9095 [eV] for P4 indicates a greater tendency of P10 to participate in the electron transfer relative to P4; hence, the first compound is a stronger radical scavenger than the second one. An identical situation takes place for the psittacofulvins derivatives, for which IP(POOHx) < IP(Px) < IP(Dx) for x = 6,7,8,9, clearly indicating that di-aldehydes D6–9 are more reactive than the parent compounds.

Important information regarding the chemical activity of the molecules considered is provided by the descriptors characterizing the electro-donating ω^−^ and electro-accepting ω^+^ power of a radical scavenger. For P4 (P10), ω^−^ = 8.1737 (10.5251) [eV] indicates that the antioxidant activity of psittacofulvins diminishes with the number of double bonds, in contradistinction to their antireductant activity, which increases according to ω^+^ = 3.8620 (6.4568) [eV] for P4 (P10). These conclusions are supported by the acceptance Ra and donation Rd indexes that take the values Ra = 1.1352 (1.8980) and Rd = 2.3557 (3.0333) for P4 (P10). Consequently, all psittacofulvins are more effective electron acceptors than F (Ra = 1) and slightly more effective electron donors than Na (Rd = 1), respectively. The Ra parameters calculated using M06-2X and BHandHLYP functionals exhibit values of Ra < 1, contrasting with Ra(B3LYP) > 1, thereby altering the interpretation of acceptor activity to be less than that of F. It is also worth noting that the activity of P4 is similar to that attributed to astaxanthin [11]: Ra = 0.94, Rd = 2.10, ω^+^ = 3.21, ω^−^ = 7.27, recognized as the most effective electron acceptor among the natural pigments [11]. If the increasing anti-reductivity from P4 to P10 is taken into account, one can expect all psittacofulvins to show greater activity in this respect than astaxanthin. Collecting the results from Table 2, Table 3 and Table 4, the following parameter relationships can be formulated: ω^+^(POOHx) < ω^+^(Px) < ω^+^(Dx), ω^−^(POOHx) < ω^−^(Px) < ω^−^(Dx), Ra(POOHx) < Ra(Px) < Ra(Dx), Rd(POOHx) < Rd(Px) < Rd(Dx) for x = 6, 7, 8, 9.

It is well-known [29] that the electrophilicity index ω can be related to the toxicity expressed, for example, in the lethal dose LD_50_ of the compound. In this respect, psittacofulvins and their derivatives are characterized by high ω values in the range (5.8181–8.3673) for P5–10, (6.3681−7.5613) for POOH6−9, and (8.2127−9.0879) for D6–9. A comparison with the ω values [29] for the highly toxic HCN (LD_50_ = 4.11 [mg kg^−1^], ω = 0.0982 [eV]) and its low toxic salt CuCN (LD_50_ = 1265.0 [mg kg^−1^], ω = 0.26991 [eV]) indicates a very low toxicity of the compounds under consideration. This development represents a positive advancement concerning the potential application of psittacofulvins and their derivatives as an ingredient in food products and supplements.

The analysis of the remaining parameters of chemical activity (EA, χ, η, S) reveals their expected dependence on the number of double bonds in all the molecules considered. Particularly, extending the polyene chain leads to a decrease in χ and η, whereas EA and S values are increased. This proves that the extension of the polyene chain affects an increase in (EA) the tendency to attach electrons in the process of radical neutralization and (S) the hardness of the molecule, i.e., susceptibility to deformation or polarization of the electron cloud under the influence of external factors (reagents). On the other hand, it leads to a decrease in (η) the molecular softness (correlated with S) and (χ) the electronegativity of all psittacofulvins and their derivatives.

The thermodynamic descriptors reported in Table 2, Table 3 and Table 4 reveal that the dominant free radical scavenging mechanism predicted for psittacofulvins and their derivatives in the water medium is sequential proton loss electron transfer (SPLET), as the values of PA are less than BDE (HAT) and AIP (SET-PT) for P6–10 and the derivatives considered. The only exceptions are P4 and P5. The results presented in Table 2 indicate a systematic change in the values of the thermodynamic parameters that depend on the number of double bonds in the molecule. Consequently, we observe the equalization of the BDE and PA values of P5, which results in a hybrid HAT + SPLET scavenging mechanism, and then in its change to HAT for P4. Intriguingly, this type of structure–activity relationship does not occur in the case of D6–9, for which only the AIP and PDE parameters depend on the carbon chain length, while the remaining ones do not show this property. In the case of POOH6–9, only the PA and G_acidity_ parameters are invariant with respect to the number of double bonds in the chain. The descriptor G_acidity_, related to the TMC mechanism, takes the large values of 311−325 [kcal mol^−1^] for P4−10, the intermediate value of 300 [kcal mol^−1^] for D6–9 and the low value of 286 [kcal mol^−1^] for POOH6–9 in the water medium which has been observed to enhance metal chelation. For comparison, the range of H_acidity_ is 330–349 [kcal mol^−1^] for P6 and its derivatives (Table 1) in the gas phase. The reported results indicate that POOH6–9 not only deactivates radicals but also slows down their creation by complexing transition metals that catalyze this process. The PA parameters are a convenient tool in the ranking of the antiradical power of the compounds investigated (water): PA(POOH6–9) = 44, PA (D6–9) = 57, PA(P6–10) = 75–68 [kcal mol^−1^]. Taking into account the total energy necessary to activate the two SPLET stages, the above ranking changes to: (PA + ETE)(D6–9) = 119–120 [kcal mol^−1^], (PA + ETE)(P6–10) = 120–123 [kcal mol^−1^], (PA + ETE)(POOH6–9) = 127–132 [kcal mol^−1^]. For comparison, gallic acid, one of the strongest radical scavengers, is characterized by PA = 43–50 and PA + ETE = 126 − 152 [kcal mol^−1^], depending on the position of the hydroxyl group involved in the reaction.

McGraw and Nogare [3] discovered that the pigment responsible for the plumage of parrots contains four dyes P6–9 and a fifth, more polar, lipochrome. The Raman spectroscopic studies performed by Tay et al. [4] revealed that the additional dye molecule contains a greater number of conjugated double bonds than that reported for P9. The results obtained in this study suggest that P10 (docosadecaenal) is a good candidate for the fifth component of the complex of naturally occurring psittacofulvins. This hypothesis is supported by the fact that P10 is endowed with the properties (Table 2) predicted in [3,4]; it has a greater polarity (d = 15.80 [D]) than P6–9 (d = 12.69–15.18 [D]), and a greater number of double bonds in the polyene chain, than P6–9.

Natural and synthetic psittacofulvins P4–10 and their POOH6–9 derivatives have an amphiphilic structure, i.e., they are composed of a polyene chain with lipophilic properties and an aldehyde or carboxyl group of a hydrophilic nature. This dual structure serves to validate the potential for these compounds to be utilized as surfactants in general, and emulsifiers in particular. Surfactants are recognized for their ability to form self-assembled molecular clusters, known as micelles, within a solution, whether in a water or oil phase. These micelles may adopt single-layer (lamellar), two-layer (bi-lamellar), or multilamellar structures. Among these, the two-layer class encompasses nano- and liposomes, extensively employed as carriers for bioactive compounds in medicinal and cosmetic applications. Given the amphiphilic nature of psittacofulvins and their derivatives, their utilization in the fabrication of active nanocarriers is justified. These nanocarriers not only facilitate the transportation of substances within their interior or a membrane but also, upon release, undergo disintegration and function as active compounds themselves.

Surfactants can be classified with respect to their applicability using the HLB (hydrophilic–lipophilic balance) parameter, defined in its simplest form by Griffin’s formula [30] (the Davies model is more advanced [31]):$$HLB=20\cdot \frac{m}{M}$$
in which m is the molecular mass of the hydrophilic part of the molecule, whereas M is the molecular mass of the whole molecule. Consequently, HLB takes the values in the range 0−20 and predicts a surfactant’s application as an anti-foaming agent (1−3), a w/o emulsifier (3−6), a wetting agent (7−9), or an o/w emulsifier (8−16). The calculations revealed that HLB(P4) = 3.9, HLB(P5) = 3.3, HLB(P6) = 2.9, HLB(POOH6) = 4.2, HLB(POOH7) = 3.7, HLB(POOH8) = 3.4, and HLB(POOH9) = 3.1. Therefore, these compounds can be used as bioactive emulsifiers, e.g., in cosmetics.

## 4. Conclusions

The calculated chemical and thermodynamic activity descriptors indicate that natural and synthetic psittacofulvins are strong anti-reductants, scavenging free radicals via the SPLET mechanism in hydrophilic environments and via HAT in a vacuum and in hydrophobic media. Their antiradical activity increases with the length of the carbon chain and the number of conjugated double bonds. When the chain length is reduced, the radical deactivation mechanism changes from SPLET to HAT. This effect is observed for the borderline case P5, which deactivates radicals using the HAT + SPLET hybrid scenario. Hydroxyl and dialdehyde derivatives of the natural psittacofulvins are also potent anti-reductants that scavenge radicals in the water by the SPLET mechanism, but their anti-radical activity does not depend on chain length. This information is crucial in the context of synthesizing analogues that may be based on the short-chain derivatives. The antiradical and antimicrobial activity of the natural psittacofulvins and their short-chain derivatives enable their use as multifunctional components of cosmetics, supplements, and medicines [32], in which they may play a triple role as preservatives, anti-radicals and dyes. Due to the expected low toxicity of all compounds considered, their application in food production [33] is also a possibility. Moreover, the amphiphilic nature of psittacofulvins and their derivatives justifies their utilization in the fabrication of active nanocarriers in medicinal and cosmetic applications.

## Figures and Tables

**Figure 1 molecules-29-02760-f001:**
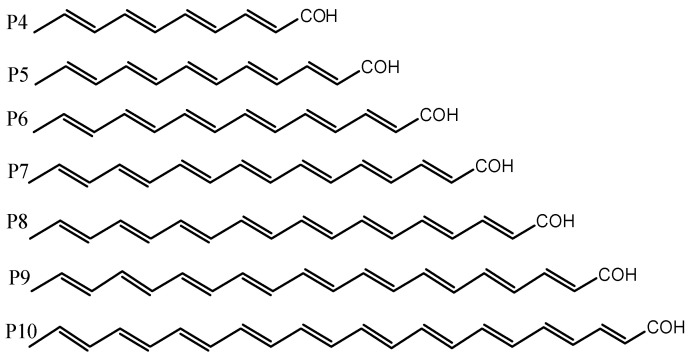
Psittacofulvins responsible for the color of parrot feathers (P6–9), the hypothetical dye component P10 (docosadecaenal), and the P4 (decatetraenal) and P5 (dodecapentaenal) derivatives not identified in the pigment complex.

**Figure 2 molecules-29-02760-f002:**
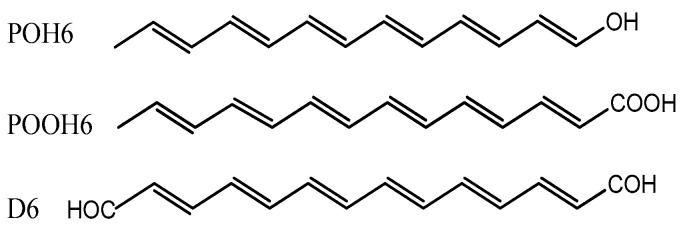
Hydroxyl (POH6—tridecahexaen-1-ol), carboxyl (POOH6—tetradecahexaenoic acid), and dialdehyde (D6—tetradecahexaenediol) derivatives of the psittacofulvin P6. The calculations performed in this study also include POOH7-9 (hexadecaheptaenoic, octadecaoctaenoic and icosanonaenoic acids) and D7–9 (hexadecaheptaenediol, octadecaoctaenediol, and icosanonaenedial) derivatives of P7–9.

**Figure 3 molecules-29-02760-f003:**
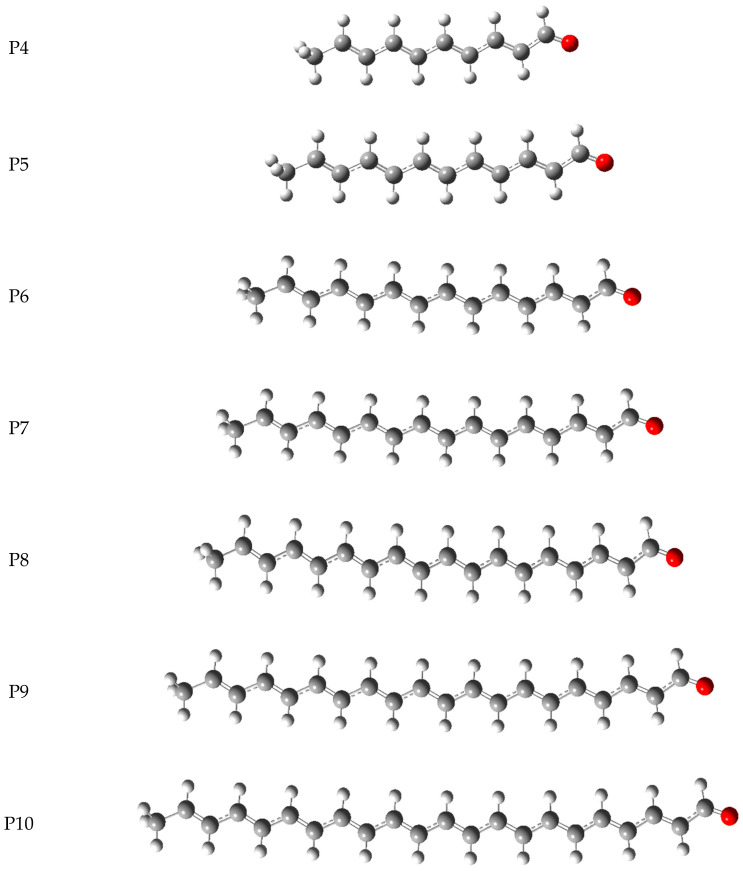
The optimized geometries of the naturally occurring psittacofulvins P6–9 and their derivatives P4, P5, P10 in the water medium evaluated by the DFT method at the B3LYP/QZVP theory level, using the C-PCM solvation model.

**Figure 4 molecules-29-02760-f004:**
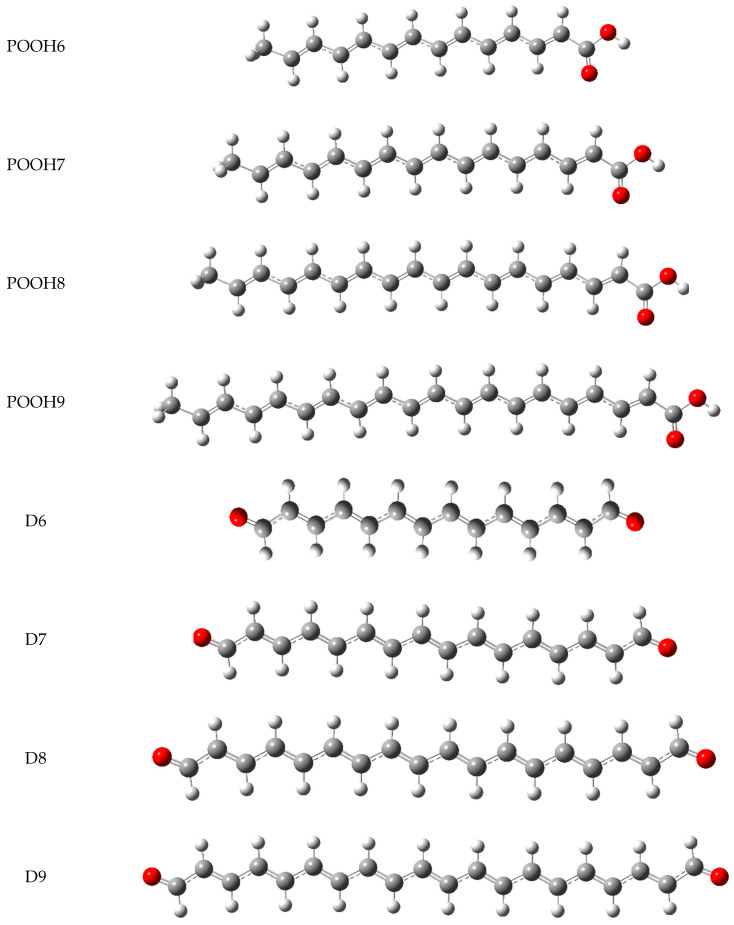
The optimized geometries of the carboxyl POOH6–9 and dialdehyde D6–9 derivatives of the psittacofulvins P6–9 (Figure 3) in the water, evaluated by the DFT method at the B3LYP/QZVP theory level and the C-PCM solvation model.

**Table 1 molecules-29-02760-t001:** The thermodynamic descriptors [kcal mol^−1^] characterizing psittacofulvin P6 and its derivatives POH6, POOH6, and D6, determined at the DFT/B3LYP/QZVP level of the theory in vacuum, water, and benzene, using the C-PCM solvation model.

Descriptor	P6	POH6	POOH6	D6
Medium	Vacuum			
BDE	74.43	100.47	102.89	72.13
PA	350.94	360.43	340.43	331.55
ETE	38.00	54.55	76.97	55.09
AIP	156.11	146.03	153.92	168.00
PDE	232.82	268.95	263.47	218.64
H_acidity_	349.46	358.95	338.95	330.31
Mechanism	HAT	HAT	HAT	HAT
Medium	Water			
BDE	76.56	99.17	86.15	73.67
PA	75.09	67.64	44.10	57.34
ETE	47.57	77.63	88.15	62.43
PA + ETE	122.66	145.27	132.25	119.77
AIP	95.28	89.55	94.58	105.64
PDE	27.38	55.71	37.66	14.13
G_acidity_	317.98	311.24	286.42	300.59
Mechanism	SPLET	SPLET	SPLET	SPLET
Medium	Benzene			
BDE	77.39	102.13	99.08	74.85
PA	119.50	122.35	99.33	101.26
ETE	55.52	77.41	97.38	71.21
AIP	93.79	126.66	133.18	144.79
PDE	81.23	73.09	63.53	27.69
Gacidity	331.68	335.08	310.94	313.85
Mechanism	HAT	HAT	HAT + SPLET	HAT

**Table 2 molecules-29-02760-t002:** The thermodynamic [kcal mol^−1^] descriptors of the natural P6–9 and projected P4, P5, P10 psittacofulvins, determined at the DFT/B3LYP/QZVP theory level in the water medium, using the C-PCM solvation model. The chemical activity descriptors [eV] are calculated from the energies E_LUMO_ = −EA, E_HOMO_ = −IP. The preferred mechanism of the radical scavenging is specified.

Descriptor	P4	P5	P6	P7	P8	P9	P10
BDE	79.24	77.70	76.56	75.69	74.99	74.41	73.92
PA	83.02	78.45	75.09	72.51	70.47	68.85	67.50
ETE	42.32	45.35	47.57	49.27	50.60	51.66	52.52
PA + ETE	125.34	123.80	122.66	121.78	121.07	120.51	120.02
AIP	105.22	99.50	95.28	92.04	89.50	87.43	85.72
PDE	20.11	24.29	27.38	29.74	31.59	33.08	34.40
G_acidity_	325.45	321.16	317.98	315.59	313.72	312.26	311.11
d ^[a]^	10.37	11.61	12.69	13.64	14.47	15.18	15.80
Mechanism	HAT	HAT-SPLET	SPLET	SPLET	SPLET	SPLET	SPLET
EA	2.7141	2.8229	2.9029	2.9633	3.0112	3.0488	3.0792
IP	5.9095	5.6611	5.4771	5.3342	5.2235	5.1334	5.0572
ΔE	3.1954	2.8381	2.5742	2.3709	2.2123	2.0847	1.9779
η	1.5977	1.4191	1.2871	1.1855	1.1061	1.0423	0.9889
S	0.3129	0.3523	0.3885	0.4218	0.4520	0.4797	0.5056
χ = −μ	4.3118	4.2420	4.1900	4.1488	4.1174	4.0911	4.0682
ω	5.8181	6.3402	6.8201	7.2598	7.6629	8.0287	8.3673
ω^+^	3.8620	4.3966	4.8859	5.3336	5.7425	6.1134	6.4568
ω^−^	8.1737	8.6386	9.0760	9.4824	9.8599	10.2045	10.5251
Ra ^[b]^	1.1352	1.2924	1.4362	1.5678	1.6880	1.7970	1.8980
Rd ^[b]^	2.3557	2.4896	2.6157	2.7328	2.8416	2.9409	3.0333

[a] Dipole moment in Debye unit [D]. [b] Dimensionless parameter.

**Table 3 molecules-29-02760-t003:** The thermodynamic [kcal mol^−1^] descriptors of the carboxylic derivatives POOH6−7 of psittacofulvins P6–9 determined at DFT/B3LYP/QZVP theory level in the water medium, using the C-PCM solvation model. The chemical activity descriptors [eV] are calculated from the energies E_LUMO_ = −EA, E_HOMO_ = −IP. The preferred mechanism of the radical scavenging is specified.

Descriptor	POOH6	POOH7	POOH8	POOH9
BDE	86.15	84.23	82.59	81.16
PA	44.10	44.11	44.12	44.13
ETE	88.15	86.22	84.57	83.13
PA + ETE	132.25	130.33	128.69	127.26
AIP	94.58	91.43	88.94	86.93
PDE	37.66	38.90	39.75	40.33
G_acidity_	286.42	286.49	286.50	286.47
Mechanism	SPLET	SPLET	SPLET	SPLET
EA	2.7856	2.8572	2.9133	2.9581
IP	5.4453	5.3073	5.1987	5.1108
ΔE	2.6596	2.4501	2.2855	2.1527
η	1.3298	1.2251	1.1427	1.0763
S	0.3760	0.4081	0.4375	0.4645
χ = −μ	4.1155	4.0823	4.0560	4.0345
ω	6.3681	6.8016	7.1981	7.5613
ω^+^	4.4766	4.9136	5.3129	5.6786
ω^−^	8.5921	8.9959	9.3689	9.7131
Ra ^[a]^	1.3159	1.4444	1.5617	1.6692
Rd ^[a]^	2.4762	2.5926	2.7001	2.7993

[a] Dimensionless parameter.

**Table 4 molecules-29-02760-t004:** The thermodynamic [kcal mol^−1^] descriptors of the dialdehyde derivatives D6–9 of psittacofulvins P6–9 determined in water at the DFT/B3LYP/QZVP theory level, using the C-PCM solvation model. The chemical activity descriptors [eV] are calculated from the energies E_LUMO_ = −EA, E_HOMO_ = −IP. The preferred mechanism of the radical scavenging is specified.

Descriptor	D6	D7	D8	D9
BDE	73.67	73.42	73.17	72.95
PA	57.34	57.04	56.80	56.60
ETE	62.43	62.47	62.47	62.44
PA + ETE	119.77	119.51	119.27	119.04
AIP	105.64	99.49	96.08	93.29
PDE	14.13	20.02	23.19	25.75
G_acidity_	300.59	300.47	300.33	300.23
Mechanism	SPLET	SPLET	SPLET	SPLET
EA	3.2975	3.3007	3.3032	3.3051
IP	5.8379	5.6510	5.5019	5.3802
ΔE	2.5405	2.3502	2.1987	2.0751
η	1.2702	1.1751	1.0993	1.0376
S	0.3936	0.4255	0.4548	0.4819
χ = −μ	4.5677	4.4759	4.4025	4.3427
ω	8.2127	8.5239	8.8154	9.0879
ω^+^	6.0876	6.4329	6.7516	7.0463
ω^−^	10.6553	10.9087	11.1541	11.3890
Ra ^[a]^	1.7894	1.8909	1.9846	2.0712
Rd ^[a]^	3.0709	3.1439	3.2146	3.2823

[a] Dimensionless parameter.

## Data Availability

All new data created are reported in this work.

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
