# Peer review of "Theoretical Insight into Psittacofulvins and Their Derivatives"

_molecules, 2024, doi:10.3390/molecules29122760_

Round 1

Reviewer 1 Report

Comments and Suggestions for Authors

This is a complete investigation with important results. My mani concern is about the molecular orbitals an eigenvalues. They were obtained at B3LYP level and this is not correct. B3LYP is not parametrized for eigenvalues. Author could remove this section, or he/she can perform a single point calculation with another funtional to obtain the eigenvalues.

Author Response

Enclosed in the Word file.

Reviewer 2 Report

Comments and Suggestions for Authors

Dear author 

I am writing to address some concerns regarding the manuscript titled “ Psittacofulvins and their Derivatives
” While the study provides valuable insights, there are several areas that require clarification and improvement.

1- Calculated Parameters Clarity: The author carried out theoretical calculations using density functional theory (DFT). However, many of the calculated parameters remain unclear. It would greatly enhance the manuscript if the author could provide more detailed explanations or references for these parameters.

2- Descriptors Missing: Some descriptors crucial for understanding the results are not adequately discussed in the present study. It is essential to include these descriptors to ensure a comprehensive analysis.

Excited States and Time-Dependent Wave Function: To explore excited states accurately, the use of time-dependent wave functions is essential. I encourage the author to

incorporate this approach to enhance the manuscript’s quality.

Please find my criticisms and recommendations in the attached file. 

Comments on the Quality of English Language

Please find my criticisms and recommendations in the attached file. 

Author Response

Enclosed in the Word file. 

Reviewer 3 Report

Comments and Suggestions for Authors

In this work, the authors study the properties of psittacofulvins, utilizing computational methods to analyze various derivatives. The findings suggest potential applications in medicine, cosmetics, supplements, and food, highlighting their roles as preservatives, radical scavengers, and coloring agents. Additionally, the study identifies a new psittacofulvin variant with specific chemical characteristics.

This work can be interesting for the computational chemistry and the industrial community. Before the publication, I would like to ask the authors to consider the minor comments below.

1. page 3, section 2 computational details

In this work, the studied systems have long chains of conjugated double bonds, where the long-range interaction can play an important role. Can the authors discuss the effects of not adding the dispersion correction in the DFT calculations?

2. page 4

References for some exchange-correlation functionals are missing. For example, M06: Zhao, Yan, and Donald G. Truhlar. Theoretical chemistry accounts 120 (2008): 215-241. BHandHLYP: J. Chem. Phys. 98, 1372–1377 (1993).

3. page 4, line 183

“the B3LYP/C-PCM combination correctly reproduces the difference in HOMO - LUMO energies”

For the systems studied in this work, will explicitly adding solvent molecules (for example, water) lead to difference results to using the PCM model?

4. page 11, Figure 5

It should be mentioned in the context that orbital energies computed at the DFT level have errors and the band gap is largely underestimated. In addition, there is also the starting point dependence in DFT calculations.

5. page 11, Figure 5

Besides the visualization of the frontier orbitals, it is also helpful to demonstrate the Mulliken charge population and the natural bond analysis for the chemical reactivity.

Comments on the Quality of English Language

No major language or grammar problem found.

Author Response

Enclosed in the Word file. 

Round 2

Reviewer 1 Report

Comments and Suggestions for Authors

Theoretical Insight into Psittacofulvins and their Derivatives

This is an interesting investigation, but there are several aspects that must be corrected.

1-    Global descriptors should be described. Are Ip and AE vertical? Author must include the equations.

2-    HAT- which atom was removed? Do they remove one by one each H atom?

3-    The same for SPLET and SET-PT

4-    B3LYP is not a good functional for eigenvalues. HOMO and LUMO must be calculated with another functional.

5-    It is not important, neither relevant to report total energies.

6-    In tables, the definitions of the abbreviations must be included.

7-    Explain how TMC is related to metal chelation?

8-    HLB section is completely useless

Author Response

Response to the reviewer's comments is included in the attached Word file. 

Reviewer 2 Report

Comments and Suggestions for Authors

Dear Author:

I hope this email finds you well. I am writing to provide my recommendation regarding the manuscript titled “[Theoretical Insight into Psittacofulvins and their Derivatives]” submitted to The Molecules Journal.

After carefully reviewing the revised manuscript and considering the author’s responses to my comments, I am pleased to report that significant improvements have been made. The author has addressed all the concerns raised during the peer review process, resulting in a more robust and coherent manuscript.

Based on the positive response from the author and the quality of the revisions, I recommend accepting the manuscript for publication in The Molecules Journal.

Sincerely,

Shaaban A. Elroby

skamel@kau.edu.sa

Author Response

Thank you for your constructive review, which helped to improve the final version of my work. Best regards, Marcin Molski